# From Values to Tokens: An LLM-Driven Framework for Context-aware Time Series Forecasting via Symbolic Discretization

## Abstract

Time series forecasting plays a vital role in supporting decision-making across a wide range of critical applications, including energy, healthcare, and finance. Despite recent advances, forecasting accuracy remains limited due to the challenge of integrating historical numerical sequences with contextual features, which often comprise unstructured textual data. To address this challenge, we propose Token-Cast, a large language model (LLM) driven framework that leverages language-based symbolic representations as a unified intermediary for context-aware time series forecasting. Specifically, TokenCast employs a discrete tokenizer to transform continuous numerical sequences into temporal tokens, enabling structural alignment with language-based inputs. To effectively bridge the semantic gap between modalities, both temporal and contextual tokens are embedded into a shared representation space via a pre-trained LLM, further optimized with autoregressive generative objectives. Building upon this unified semantic space, the aligned LLM is subsequently fine-tuned in a supervised manner to predict future temporal tokens, which are then decoded back into the original numerical space. Extensive experiments are conducted on multiple real-world datasets, whose results reveal the performance of our framework and highlight its potential as a generative framework for multimodal time series forecasting. The code is available for further research at: https://anonymous.4open.science/r/TokenCast-8EFF.

## 1 Introduction

Time series forecasting (TSF) is critical for decision-making in domains such as energy (Das et al., 2023; Jin et al., 2024; Wang et al., 2025), healthcare (Qiu et al., 2024), and finance (Feng et al., 2019). The goal is to predict future values based on historical observations and associated contextual features. In practice, accurate forecasting requires not only modeling temporal dependencies in numerical sequences, but also understanding how they interact with external contextual factors—such as static attributes or dynamic events (Liu et al., 2024b). Fundamentally, TSF can be viewed as learning a mapping from past values and contextual features to future outcomes (Jiang et al., 2025).

To learn this mapping, researchers have proposed a comprehensive range of methods, ranging from classical statistical models to modern data-driven approaches. Traditional methods, such as ARIMA (Hyndman & Khandakar, 2008) and state-space models (Winters, 1960), rely on strong assumptions about data generation and often incorporate domain-specific priors. In contrast, recent data-driven approaches such as deep learning models aim to learn patterns directly from data without hand-crafted assumptions. Architectures based on RNNs (Lai et al., 2018), CNNs (Cheng et al., 2025b), Transformers (Zhou et al., 2022), and MLPs (Challu et al., 2023) have been widely adopted, each capturing different aspects of temporal dependencies. However, most of these models assume homogeneous numerical inputs and struggle to effectively incorporate complex contextual features, particularly those with heterogeneous modalities.

Beyond capturing temporal dependencies, there is an increasingly growing emphasis in recent research on incorporating contextual features to enhance forecasting performance (Liu et al., 2024a; Williams et al., 2024; Liu et al., 2024b). These features typically fall into two categories: dynamic exogenous variables (e.g., weather conditions, event indicators) and static attributes (e.g., product

types, patient demographics, market segments). When contextual features share the same numerical modality as the target series, they can be directly modeled as additional input channels. However, many particularly high-value contextual features, such as clinical notes, policy texts, or user logs, are expressed in unstructured textual form. This heterogeneity poses significant challenges for aligning and integrating information across modalities.

To address these challenges, some studies have explored shallow fusion strategies to incorporate contextual features. Models such as DeepAR (Salinas et al., 2020) and Temporal Fusion Transformer (TFT) (Lim et al., 2021) typically concatenate external variables with time series or introduce gating mechanisms. While offering basic integration, these methods often rely on weak alignment and struggle to capture deep semantic interactions across modalities (Liu et al., 2024e). More recently, LLMs have been introduced into time series forecasting (Sun et al., 2023; Liu et al., 2024c; Ansari et al., 2024). Methods like Time-LLM (Jin et al., 2023) inject time series features into LLMs using linear adapters (Figure 1 (a)) or soft prompts (Figure 1 (b)). Although promising, these approaches fall short in resolving the structural discrepancies between numerical sequences and unstructured contextual features. Moreover, they fail to fully leverage the generative and reasoning capabilities of LLMs, which are pretrained on large-scale corpora. This observation raises a fundamental question: *Can time series be effectively modeled in a discrete token space to unlock the full potential of LLMs?*

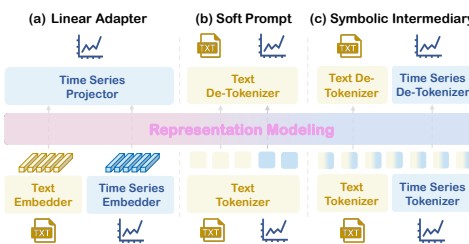

Figure 1: Methods for representation modeling of time series and contextual features: (a) linear adapter, (b) soft prompt, and (c) symbolic intermediary.

Motivated by this question, we explore a more expressive yet under-explored paradigm that formulates time series forecasting as a multimodal discrete context understanding and generation problem, powered by pre-trained LLMs, as illustrated in Figure 1 (c). The key idea is to transform continuous numerical sequences into discrete tokens and embed them into the same semantic space as contextual language inputs. This formulation enables the full use of LLMs' capabilities in semantic understanding, contextual reasoning, and autoregressive generation. However, this paradigm introduces several non-trivial challenges. First, discretizing dynamic time series is more difficult than compressing static data, as it requires preserving temporal dependencies while reducing granularity. Second, even with symbolic representations, semantic misalignment between temporal tokens and contextual features may hinder effective fusion. Finally, it remains unclear whether time series forecasting can be effectively addressed through autoregressive generation over discrete tokens—a direction still largely unexplored.

Based on the above analysis, we propose TokenCast, an LLM-driven framework for context-aware time series forecasting via symbolic discretization. TokenCast begins with a time series tokenizer that converts continuous sequences into temporal tokens, mitigating structural discrepancies across data modalities. To bridge the semantic gap, temporal and contextual tokens are jointly embedded into a shared representation space using a pre-trained LLM, optimized via an autoregressive objective while keeping the backbone frozen and tuning only the embedding layer. Building on this unified semantic space, the aligned LLM is further fine-tuned with supervised forecasting signals to enhance predictive performance. We evaluate TokenCast on diverse real-world datasets enriched with contextual features. Experimental results show that TokenCast achieves strong accuracy and generalization across domains. We also conduct comprehensive ablation and qualitative studies, offering insights into the flexibility of symbolic, LLM-based time series forecasting.

## 2 RELATED WORK

Time series forecasting (TSF) is a fundamental task across various domains. Traditional approaches typically rely on statistical assumptions such as stationarity and linearity, and often depend on hand-crafted assumptions that limit their flexibility (Holt, 2004; Kalekar et al., 2004). Alternatively, data-driven methods (Chen & Guestrin, 2016), particularly those based on deep learning, have advanced TSF by learning temporal patterns directly from data. RNN-based models (Wang et al., 2019) capture dependencies through recurrence, CNN-based models (Wang et al., 2023) enhance local pat-

tern extraction, and Transformer-based architectures (Shi et al., 2024) are well-suited for modeling long-range interactions. Furthermore, MLP-based approaches (Wang et al., 2024b) demonstrate that simple architectures can achieve competitive performance with improved computational efficiency. These models mainly focus on numerical data, with less emphasis on unstructured context.

In addition to modeling temporal dependencies, recent research increasingly emphasizes the integration of contextual features for accurate forecasting (Chang et al., 2023; Liu et al., 2024d; Hu et al., 2025). Two major lines of research have emerged in this direction. One line of research focuses on deep learning architectures that explicitly model feature interactions (Gasthaus et al., 2019). For example, TimeXer (Wang et al., 2024c) employs cross-attention mechanisms to fuse dynamic and static modalities. Another line of research leverages pre-trained LLMs for multimodal modeling (Cheng et al., 2025a; Liu et al., 2025). Some approaches, such as TEMPO (Cao et al., 2023), utilize linear adapters to project time series features into the LLM's semantic space. Others, like Promptcast (Xue & Salim, 2023), employ soft prompts to guide the frozen LLM's behavior. However, these promising approaches fail to bridge the structural gap between numerical and textual modalities.

## 3 THE PROPOSED TOKENCAST

In this section, we present the precise formal problem definition, clarify the key concepts and notations used consistently throughout the paper, and provide an overview of the TokenCast.

### 3.1 PROBLEM FORMULATION

We consider a dataset $\mathcal{D} = \{(X_i, T_i, P_i)\}_{i=1}^N$ of $N$ multimodal time series instances. For each instance, $X \in \mathbb{R}^{L \times C}$ represents the multivariate time series over $L$ time steps and $C$ channels, $T$ denotes the contextual features, and $P \in \mathbb{R}^{L_P \times C}$ is the ground-truth future sequence over a horizon $L_P$. The contextual features $T$ are tokenized to tokens $Y$ using the tokenizer of a pre-trained LLM, while the time series $X$ is converted into discrete tokens $Z_q$ via a learnable mapping $f_\theta : X \mapsto Z_q$. These two token sequences are then concatenated to form a token sequence $Z = [Z_q; Y] \in \mathcal{V}^{T'}$. We use boundary markers to delimit the temporal tokens of $\hat{Z}$. Finally, a decoding function $g_\phi : \hat{Z} \mapsto \hat{P}$ is applied to reconstruct the raw time series $\hat{P} \in \mathbb{R}^{L_P \times C}$.

### 3.2 FRAMEWORK OVERVIEW

Figure 2 illustrates the overview of the TokenCast, which consists of three main stages. The process begins with the time series tokenizer, which transforms continuous time series into a sequence of discrete tokens via a decoupled and dynamical vector quantization tokenizer. Subsequently, both the temporal and contextual tokens are then jointly processed by a pre-trained LLM, which performs cross-modality alignment under autoregressive objectives. Following this alignment, the aligned LLM is adapted to the forecasting task via generative fine-tuning, enabling token prediction. The predicted tokens are decoded to raw time series using a frozen time series de-tokenizer. The following sections elaborate on the principal stages of the TokenCast.

### 3.3 TIME SERIES DISCRETIZATION

#### 3.3.1 TIME SERIES TOKENIZER

To fully harness the generative and reasoning capabilities of language models, symbolic representation naturally arises as an effective intermediary. Accordingly, we employ time series discretization as a simple yet powerful approach to establish this bridge. It is worth noting that existing approaches, such as Symbolic Aggregate Approximation (SAX) (Lin et al., 2007), have achieved progress in time series discretization but often suffer from significant information loss due to dimensionality reduction. In contrast, reconstruction-based methods (Van Den Oord et al., 2016) map subsequences to discrete codes from a predefined codebook and achieve more precise representations through reconstruction optimization. While preserving the original information is advantageous, previous reconstruction-based methods typically encode the entire sequence, overlooking the statistical properties of time series. In the forecasting task, Reversible Instance Normalization (RevIN) (Kim et al.,

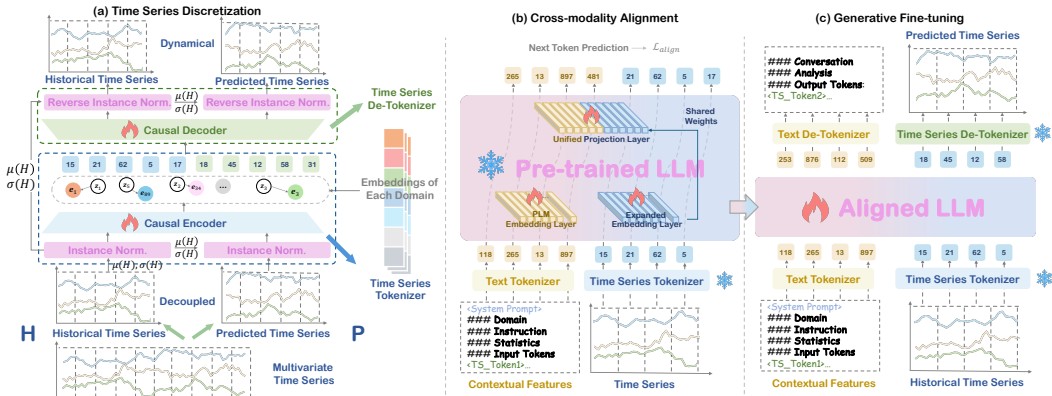

Figure 2: Overview of the framework for context-aware time series forecasting: (a) time series tokenizer to address the structural differences between modalities, (b) cross-modality alignment with an autoregressive objective to bridge the modalities, and (c) generative fine-tuning and context-aware forecasting through time series decoding for horizon prediction.

2021) is widely used, yet its normalization and denormalization steps risk leaking future information. To overcome this limitation, we propose a decoupled and dynamic tokenizer.

As illustrated in Figure 2 (a), similar to the forecasting phase, we divide the multivariate time series into a historical time series $H \in \mathbb{R}^{L_H \times C}$ and a predicted time series $P \in \mathbb{R}^{L_P \times C}$, which can be formally represented as $X = [H; P] \in \mathbb{R}^{L \times C}$. The process begins with a reversible instance normalization (RIN) layer. We compute the mean $\mu(H)$ and standard deviation $\sigma(H)$ solely from the historical time series $H$, and apply them to normalize the time series $X$, thereby preventing future information leakage. These statistics are retained for inverse transformation during decoding. Instead of employing separate encoders, we adopt a shared encoder, which facilitates the joint modeling of both local and global information. The normalized time series is then passed through a causal encoder $f_{\text{enc}}$, yielding a sequence of continuous latent representations $Z = f_{\text{enc}}(X) \in \mathbb{R}^{T \times d}$, where $T$ is the number of latent vectors and $d$ is the feature dimension. To discretize the latent representations, we apply a vector quantization layer. For domain $i$, a learnable codebook $C_i = \{e_{i,k}\}_{k=1}^{K} \subset \mathbb{R}^d$ is maintained, containing $K$ embedding vectors. Each latent vector $z_t \in \mathbb{R}^d$ is mapped to its nearest neighbor in the codebook as $z_t^q = e_{i,k^*}$, where $k^* = \arg\min_k \|z_t - e_{i,k}\|_2^2$. The output of this layer is a quantized sequence $Z_q = (z_1^q, \ldots, z_T^q)$, and the corresponding sequence of indices $\{k^*\}$ serves as the discrete tokens for downstream modeling. These tokens are subsequently decoded by a shared causal decoder $f_{\text{dec}}$, rather than by separate decoders, which ensures consistent reconstruction and enables the predicted part to dynamically exploit richer historical features. Then, the final reconstruction $\hat{X}$ is obtained by applying the inverse RIN operation using the stored statistics $\mu(H)$ and $\sigma(H)$, i.e., $\hat{X} = f_{\text{denorm}}(f_{\text{dec}}(Z_q))$.

### 3.3.2 TRAINING OBJECTIVE

The tokenizer is optimized by minimizing the objective function defined as follows:

$$\mathcal{L} = \mathcal{L}_{\text{recon}} + \beta \left( \mathcal{L}_{\text{commit}} + \mathcal{L}_{\text{codebook}} \right) + \gamma \mathcal{L}_{\text{diversity}}, \tag{1}$$

where $\mathcal{L}_{\text{recon}} = \|\hat{X} - X\|_2^2$ is the reconstruction loss that optimizes both the encoder and decoder. Due to the non-differentiability of the $\arg\min$ operation in quantization, we employ the straight-through estimator (STE) during backpropagation. To train the vector quantizer, we include: $\mathcal{L}_{\text{codebook}} = \|\text{sg}[Z] - Z_q\|_2^2$, $\mathcal{L}_{\text{commit}} = \|Z - \text{sg}[Z_q]\|_2^2$, where $\text{sg}[\cdot]$ denotes the stop-gradient operator, which prevents gradients from flowing into its argument during backpropagation. To avoid codebook collapse and promote diverse usage of codebook entries, we further add a diversity loss $\mathcal{L}_{\text{diversity}} = \frac{1}{N} \sum_{i=1}^{N} \frac{1}{d_i + \epsilon}$, where $d_i = \min_{j \neq i} \|e_i - e_j\|_2$ denotes the nearest-neighbor distance between codebook embeddings. This penalty discourages vectors from clustering too closely and encourages more uniform utilization of the codebook.

### 3.4 PRE-TRAINED LLM BACKBONE FORMULATION

Following the discretization of time series into discrete tokens, the next challenge is to model the complex dependencies embedded in these sequences. While architectures like TCNs or Transformers can be trained from scratch, we argue that a pre-trained LLM serves as a more effective backbone. This is supported by two observations: (1) a pre-trained LLM possesses strong semantic understanding and contextual reasoning capabilities acquired from large-scale corpora, and (2) the structure of discrete time series tokens closely resembles that of language tokens (Zhao et al., 2023). By casting forecasting as a generative task, we directly leverage the LLM's autoregressive generation ability. To guide LLM reasoning and incorporate contextual features, we employ a structured prompt template, as shown in Figure 2 (b). This prompt template consists of four essential components: domain knowledge, task instructions, statistical properties, and discrete time series tokens. This design ensures token-level consistency with language tokens and introduces task-specific descriptions alongside statistical attributes, enabling the LLM to perform instruction-driven generation.

### 3.5 CROSS-MODALITY ALIGNMENT OF TIME SERIES AND CONTEXTUAL FEATURES

While discretization aligns time series structurally with language tokens, a semantic gap remains between time series and contextual features. Existing methods often introduce projection modules (e.g., MLPs) to map time series into the LLM's latent space for fusion (Liu et al., 2025). Although effective in downstream tasks, these strategies rely on external transformation modules for alignment, which bypass the language model's native vocabulary modeling mechanism. To this end, we implement a more explicit vocabulary-level alignment strategy. As illustrated in Figure 2 (b), we construct a unified vocabulary by directly appending $K$ temporal tokens and $S$ task-specific special tokens to the original vocabulary $V_{\mathrm{orig}}$ of the pre-trained LLM, forming an extended vocabulary $V$. Correspondingly, a shared embedding matrix $E \in \mathbb{R}^{|V| \times d}$ is used to encode all tokens, regardless of their modality origin. This unified embedding mechanism enables seamless fusion of time series and contextual features while maintaining alignment with the pre-trained model. To ensure distributional alignment with pretrained embeddings for fine-tuning, the embedding of the newly introduced time series tokens is initialized by sampling from a multivariate gaussian distribution defined by the mean $\mu$ and covariance $\Sigma$ of the original word embeddings. Then, temporal tokens $Z_q$ and contextual tokens $Y$ are concatenated at the token level and jointly transformed into embeddings via the shared embedding layer: $E([Z_q, Y]) = [E(z_1), \ldots, E(z_n), E(y_1), \ldots, E(y_m)]$, where $E$ denotes the unified embedding matrix. This unified embedding process enables the LLM to reason over concatenated sequences without requiring architectural modification.

To optimize cross-modality token representations within the shared embedding space, we adopt an autoregressive training objective. Specifically, we freeze all parameters of the pre-trained LLM and update only the shared embedding matrix $E$, which is responsible for encoding both temporal and contextual tokens. Given a concatenated token sequence $[Z_q, Y]$, the training objective is formulated as a next-token prediction task over the combined sequence:

$$\mathcal{L}_{\mathrm{align}} = -\sum_{t=1}^{T} \log p(z_t \mid z_1, \ldots, z_{t-1}; E), \tag{2}$$

where $z_t \in V$ denotes the $t$-th token in the sequence, and $p(\cdot)$ is the conditional probability predicted by the frozen language model given the embedding vectors from $E$.

### 3.6 GENERATIVE FINE-TUNING AND CONTEXT-AWARE TIME SERIES FORECASTING

We now detail the procedure for adapting the aligned LLM for forecasting tasks. As illustrated in Figure 2 (c), we employ a generative fine-tuning strategy to specialize the model for context-aware time series forecasting. This process consists of two primary stages: (1) structured prompt-based generative fine-tuning; and (2) context-aware time Series forecasting with token-based decoding. In the first stage, prompt-based generative fine-tuning is introduced to explicitly transfer the pretrained language modeling capability into the forecasting domain. Instead of relying on external mapping modules, generative fine-tuning directly formulates forecasting as a generation task, where the model is supervised to output both natural language reasoning and sequences of future time series tokens. This paradigm fosters a fast-thinking behavior: by optimizing an autoregressive objective

against ground-truth structured responses, the model learns to rapidly recognize patterns, associate contextual features with temporal dynamics, and produce coherent outputs without engaging in deep deliberation. As a result, the aligned LLM acquires the ability to generate fluent and context-aware predictions. In the second stage, the fine-tuned model is utilized for context-aware forecasting and decoding. During inference, the model receives a prompt with historical data and contextual features, and autoregressively generates a complete response. The key component of this generated output is the sequence of discrete tokens, which represents the model's prediction of future time series values. To translate this symbolic representation back into a continuous predicted time series, these tokens are processed by a frozen time series de-tokenizer. We use boundary markers to delimit the temporal tokens within the generated sequence. This procedure leverages the LLM's reasoning capacity, enabling reliable forecasting grounded in the contextual feature.

## 4 EXPERIMENTS

In this section, we conduct comprehensive experiments to evaluate our TokenCast's performance on diverse, representative, and challenging real-world datasets enriched with contextual features for time series forecasting. Additionally, we perform ablation studies and exploration analysis.

### 4.1 EXPERIMENTAL SETUP

#### 4.1.1 DATASETS

As shown in Table 3, we evaluate our framework on six real-world datasets from diverse domains enriched with contextual features: **Economic** (McCracken & Ng, 2016), **Health** (Panagopoulos et al., 2021), **Web** (Gasthaus et al., 2019), two subsets of **Stock** data (Feng et al., 2019) and **Nature** (Poyatos et al., 2020). These datasets, spanning various temporal patterns and contextual dependencies, collectively serve as a comprehensive bench-

| Dataset | Domain | Frequency | Length | Variables |
|---------|--------|-----------|--------|-----------|
| Economic | Economic | 1 day | 728 | 107 |
| Health | Health | 1 day | 1,392 | 948 |
| Web | Web | 1 day | 792 | 2,000 |
| Stock-NY | Stock | 1 day | 1,243 | 5 |
| Stock-NA | Stock | 1 day | 1,244 | 5 |
| Nature | Nature | 30 mins | 19,934 | 11 |

Figure 3: Diverse real-world datasets from various domains and with distinct characteristics.

mark for context-aware forecasting. Data preparation involves imputing missing values and applying z-score normalization to all datasets, thereby ensuring stable convergence and fair comparability. A detailed description of the datasets, preprocessing procedures, and additional implementation details is provided in the Appendix A for clarity, transparency, and reproducibility.

#### 4.1.2 BASELINES

We compare our proposed framework against eight strong baselines, grouped into four representative categories for comprehensive evaluation. For LLM-based models, we include Time-LLM (Jin et al., 2023) and GPT4TS (Zhou et al., 2023), which adapt pre-trained LLMs for time series forecasting using modality-aware prompting and reprogramming. In the self-supervised frameworks category, we evaluate TimeDART (Wang et al., 2024a) and SimMTM (Dong et al., 2023). These unimodal pretraining methods leverage self-supervised objectives to enhance time series representation learning. Additionally, we include Transformer-based methods like Autoformer (Wu et al., 2021) and Crossformer (Zhang & Yan, 2023). Finally, we consider the MLP-based method DLinear (Zeng et al., 2023). Further details are provided in the Appendix B.1.

#### 4.1.3 IMPLEMENTATION DETAILS

For each baseline, we search over multiple input lengths and report the best performance to avoid underestimating its capability. The historical length is set to $L = 96$ for the Nature dataset and $L = 36$ for the other five datasets, based on the data volume and temporal resolution. The forecasting horizons are set to $\{24, 48, 96, 192\}$ for Nature and $\{24, 36, 48, 60\}$ for the other dataset. We adopt two widely used evaluation metrics in time series forecasting: mean absolute error (MAE) and mean squared error (MSE). For exploratory analysis, we use 96-to-24 on Nature and 36-to-24 on the

Table 1: All reported results are averages over four horizons and three trials on various context-rich benchmark datasets. Lower values indicate better performance. The best results are highlighted in **bold**, and the second-best are underlined.

| Model | TokenCast | | Time-LLM | | GPT4TS | | TimeDART | | SimMTM | | Crossformer | | Autoformer | | DLinear | |
|---|---|---|---|---|---|---|---|---|---|---|---|---|---|---|---|---|
| Metrics | MSE | MAE | MSE | MAE | MSE | MAE | MSE | MAE | MSE | MAE | MSE | MAE | MSE | MAE | MSE | MAE |
| Economic | **68.911** | **1.701** | 81.542 | 1.760 | 85.947 | 1.716 | 86.029 | 1.771 | 90.351 | 1.672 | 406.418 | 4.074 | 116.745 | 2.088 | 122.216 | 2.070 |
| Health | **2.525** | **0.081** | 2.823 | 0.104 | 2.565 | 0.083 | 2.623 | 0.088 | 2.720 | 0.088 | 1644.745 | 2.504 | 2.617 | 0.265 | 28.587 | 0.455 |
| Web | **497.410** | **1.246** | 557.833 | 1.751 | 540.492 | 1.458 | 773.635 | 1.369 | 847.649 | 1.327 | 698.316 | 1.963 | 722.506 | 3.303 | 632.301 | 1.398 |
| Stock-NY | **0.482** | **0.455** | 0.662 | 0.510 | 0.638 | 0.502 | 0.776 | 0.606 | 0.613 | 0.585 | 1.111 | 0.912 | 0.676 | 0.573 | 0.999 | 0.754 |
| Stock-NA | **1.134** | **0.780** | 1.200 | 0.925 | 1.272 | 0.880 | 1.409 | 0.883 | 1.343 | 0.834 | 1.913 | 1.053 | 1.558 | 0.914 | 1.710 | 0.958 |
| Nature | 0.269 | 0.297 | 0.258 | 0.283 | 0.274 | 0.299 | **0.243** | **0.273** | 0.259 | 0.286 | 0.735 | 0.511 | 0.508 | 0.481 | 0.369 | 0.436 |
| 1st Count | **5** | **5** | 0 | 0 | 0 | 0 | 1 | 1 | 0 | 0 | 0 | 0 | 0 | 0 | 0 | 0 |

other datasets. Complete results for the main experiments, ablation studies, and exploratory analysis are included in the Appendix C. All experiments are implemented in PyTorch and conducted on a distributed setup with 8 NVIDIA A100 GPUs.

## 4.2 FORECASTING PERFORMANCE ANALYSIS

Table 1 comprehensively compares forecasting performance across six benchmark datasets. Token-Cast demonstrates superior performance in most scenarios, further confirming previous empirical findings (Zhou et al., 2023) that no single model performs best across all settings. This performance highlights its adaptability across most diverse forecasting domains. Notably, LLM-based baselines like Time-LLM also show competitive results, particularly on context-rich datasets such as Economic and Stock-NY. This further validates the potential of leveraging large language models in time series forecasting. However, these models often lack the structural alignment mechanisms introduced by our framework, limiting their consistent performance. Conventional baselines such as TimeDART perform well on datasets with strong periodicity and weak contextual dependence (e.g., Nature), but their performance drops significantly on complex datasets rich in contextual features (e.g., Economic and Web). This contrast underscores the importance of contextual feature modeling and cross-modal interaction. In summary, our framework delivers state-of-the-art results with high consistency. This is attributed to its core design: discretizing time series into discrete tokens and aligning them with contextual features. This unified token-based paradigm effectively captures multimodal dependencies and addresses real-world context-aware time series forecasting challenges.

## 4.3 ABLATION STUDIES

### 4.3.1 ABLATION ON ALIGNMENT AND FINE-TUNING

We conduct the ablation study on two crucial training steps: cross-modality alignment and generative fine-tuning. The results in Figure 4 (left) clearly demonstrate their indispensable contribution to the overall framework. The cross-modality alignment stage consistently achieves lower MSE scores across all datasets. Without alignment, contextual features risk being misinterpreted by the time series backbone, leading to suboptimal forecasts. This highlights its role in bridging structural and semantic discrepancies between time series and contextual features, thus facilitating meaningful feature interaction. Meanwhile, the generative fine-tuning stage further enhances performance, with notable improvements on complex datasets such as Stock-NA. These findings emphasize the necessity of both alignment and fine-tuning in enabling reliable forecasting.

### 4.3.2 ABLATION ON MULTIMODAL CONTRIBUTIONS

Figure 4 (right) examines the impact of multimodal context by selectively removing different types of contextual features. The results demonstrate that both general information (e.g., domain knowledge and task instructions) and local information (e.g., event-specific details) make substantial contributions to forecasting accuracy across datasets. Removing either type consistently degrades performance, with the absence of local information showing particularly severe effects on datasets characterized by dynamic and non-stationary patterns. Meanwhile, excluding textual context leads to the most significant accuracy drop, underscoring the critical role of text in capturing domain

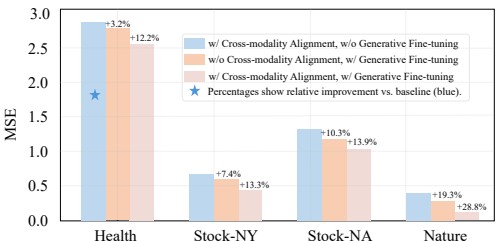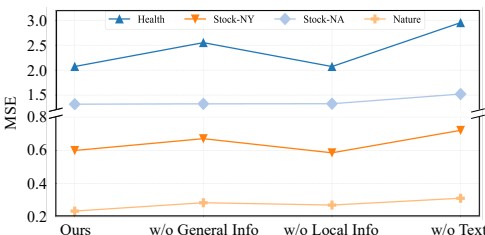

Figure 4: Ablation studies. **(Left)** Ablation study on the effects of cross-modality alignment and generative fine-tuning across multiple datasets. **(Right)** Ablation study on multiple datasets on the contribution of multimodal context in time series forecasting.

knowledge and high-level semantics. These findings highlight the complementary nature of different contextual modalities: while general information provides broad background knowledge, local information introduces fine-grained event-level signals.

Table 2: Study on the number of tokens in the codebook across multiple datasets. We report predicted reconstructed MSE (Recon.), downstream MSE, and downstream MAE.

| Dataset | Economic | | | Health | | | Web | | | Stock-NY | | | Stock-NA | | | Nature | | |
|---|---|---|---|---|---|---|---|---|---|---|---|---|---|---|---|---|---|---|
| Metrics | Recon. | MSE | MAE | Recon. | MSE | MAE | Recon. | MSE | MAE | Recon. | MSE | MAE | Recon. | MSE | MAE | Recon. | MSE | MAE |
| 32 | 190.371 | 50.234 | 1.372 | 207.459 | **1.772** | **0.065** | 731.474 | 451.827 | **1.165** | 0.569 | **0.325** | **0.377** | 0.244 | 0.794 | 0.636 | 0.134 | 0.233 | 0.281 |
| 64 | **141.852** | **37.699** | 1.293 | 101.652 | 2.714 | 0.072 | **664.501** | 529.401 | 1.228 | 0.573 | 0.339 | 0.381 | 0.213 | 0.690 | 0.616 | 0.158 | 0.241 | 0.296 |
| 128 | 170.630 | 39.379 | **1.251** | 186.619 | 2.622 | 0.070 | 3924.953 | 1743.889 | 1.539 | **0.518** | 0.730 | 0.604 | **0.205** | 0.671 | 0.600 | **0.104** | **0.203** | **0.265** |
| 256 | 191.937 | 39.309 | 1.339 | **69.035** | 2.413 | 0.070 | 5062.452 | 899.202 | 1.385 | 0.572 | 0.384 | 0.424 | 0.209 | **0.646** | **0.593** | 0.114 | 0.248 | 0.288 |

## 4.4 EXPLORATION ANALYSIS

### 4.4.1 CODEBOOK SIZE

We investigate the effect of codebook size on model performance, as summarized in Table 2. The results show that a moderate size of 128 achieves state-of-the-art performance on challenging datasets such as Nature and Stock-NA, while a smaller size of 64 yields the best results on the Economic dataset. In contrast, both overly small (32) and overly large (256) codebooks degrade performance, indicating that simply increasing token granularity does not necessarily benefit forecasting. Overall, an appropriately balanced codebook size provides a better trade-off between reconstruction fidelity and downstream forecasting accuracy.

Table 3: Performance comparison of different backbone models and their variants (base/instruct) across varying model scales and multiple datasets.

| Dataset | Economic | | Health | | Web | | Stock-NY | | Stock-NA | | Nature | |
|---|---|---|---|---|---|---|---|---|---|---|---|---|
| Metrics | MSE | MAE | MSE | MAE | MSE | MAE | MSE | MAE | MSE | MAE | MSE | MAE |
| Qwen2.5-0.5B-base | 37.164 | 1.301 | 2.492 | **0.068** | 586.793 | **1.271** | **0.297** | **0.355** | **0.668** | **0.605** | **0.180** | **0.246** |
| Qwen2.5-0.5B-inst. | **36.744** | 1.299 | 2.493 | **0.068** | **586.780** | **1.271** | 0.353 | 0.391 | 0.695 | 0.614 | 0.187 | 0.253 |
| Qwen2.5-1.5B-inst. | 38.549 | **1.283** | 2.471 | 0.069 | 589.843 | 1.273 | 0.329 | 0.372 | 0.722 | 0.611 | 0.229 | 0.270 |
| Qwen3-0.6B-inst. | 39.629 | 1.315 | **2.320** | **0.068** | 588.379 | 1.272 | 0.405 | 0.417 | 0.936 | 0.715 | 0.236 | 0.281 |

### 4.4.2 LLM BACKBONE

We evaluate four LLM backbones to identify the optimal architecture for our forecasting framework. As summarized in Table 3, the Qwen2.5-0.5B-base models consistently demonstrate superior performance. Specifically, the base version achieves state-of-the-art results on the Nature and Stock-NA datasets, while the instruct-tuned version excels on the more complex Economic dataset. Interestingly, larger models like Qwen2.5-1.5B-instruct fail to yield further gains and often underperform. This suggests that for our tasks, simply scaling up model size is not beneficial. Instead, the 0.5B models strike a balance between representational capacity and generalization.

Table 4: Study on different initialization methods on the embedding layer. We compare mean initialization, codebook sampling, and random initialization.

| Dataset | Economic | | Health | | Web | | Stock-NY | | Stock-NA | | Nature | |
|---|---|---|---|---|---|---|---|---|---|---|---|---|
| Metrics | MSE | MAE | MSE | MAE | MSE | MAE | MSE | MAE | MSE | MAE | MSE | MAE |
| Mean Initialization | **36.744** | 1.299 | **2.357** | **0.068** | **585.064** | **1.256** | **0.319** | **0.371** | 0.695 | 0.614 | **0.187** | **0.253** |
| Codebook Sampling | 39.680 | **1.261** | 2.574 | **0.068** | 585.665 | 1.265 | 0.337 | 0.380 | **0.667** | **0.602** | 0.224 | 0.264 |
| Random Initialization | **36.744** | 1.299 | 2.493 | **0.068** | 586.780 | 1.271 | 0.353 | 0.391 | 1.101 | 0.725 | 0.189 | 0.256 |

### 4.4.3 EMBEDDING LAYER INITIALIZATION

We investigate three initialization strategies for our model's embedding layer to identify the most effective approach. As shown in Table 4, mean initialization consistently provides the most reliable performance. Specifically, it achieves the best results on the Nature and Economic datasets. While word initialization is superior on the Stock-NA dataset, its performance is less consistent across other domains. Notably, standard random initialization suffers a significant performance degradation on Stock-NA, highlighting its instability. These findings suggest that initializing embeddings with meaningful prior information provides a better starting point for optimization. Therefore, we adopt mean initialization as the default.

### 4.4.4 QUALITATIVE ANALYSIS OF TOKENIZATION

To evaluate our discretization module, we analyze the Nature dataset from three perspectives, as shown in Figure 5. The token usage heatmap (left) shows that all 64 tokens are activated, mitigating codebook collapse and capturing diverse temporal structures. The codebook clustering visualization (middle) illustrates that tokens form coherent groups in the latent space, indicating that the learned vocabulary preserves structural relationships among temporal patterns. The dynamic reconstruction results (right) highlight the tokenizer's adaptive decoding property: the same token id (e.g., ID = 18) can produce different decoded segments depending on context, ensuring alignment with the original sequences. Overall, these findings confirm that our discretization process learns a diverse, semantically organized vocabulary while supporting context-aware decoding for forecasting.

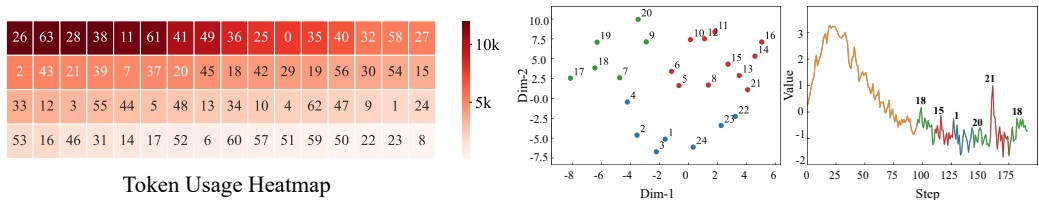

Figure 5: Visualization of tokenizer behavior on the Nature dataset. **(Left)** Token usage heatmap of the 64-token vocabulary. **(Middle)** Codebook clustering in the latent space. **(Right)** Dynamic reconstruction illustrating dynamic decoding.

## 5 CONCLUSION

We proposed TokenCast, a context-aware time series prediction framework based on a pretrained LLM. This approach first converts a continuous time series into discrete tokens. Leveraging a pretrained LLM, it aligns the temporal and contextual tokens through an autoregressive objective, achieving unified modeling of both modalities. The model is then further fine-tuned to generate future token sequences. We evaluate TokenCast on multiple real-world datasets rich in contextual information. Experimental results demonstrate that TokenCast achieves superior accuracy. We also conduct comprehensive ablation experiments and qualitative analysis to validate the framework's adaptability and flexibility for symbolic, LLM-driven time series prediction. Looking ahead, we believe that leveraging language as a symbolic intermediary will have the potential to advance time series prediction towards a multimodal and multi-task level.

## ETHICS STATEMENT

This work adheres to the ICLR Code of Ethics. Our study focuses on methodological advances in time series forecasting and does not involve human subjects, personal information, or any sensitive data. All datasets used in our experiments are publicly available and widely adopted in prior research. We strictly follow the respective dataset licenses and provide detailed preprocessing steps in the supplementary material to ensure transparency. The proposed methods are intended for scientific and practical forecasting applications, and we do not anticipate direct harmful impacts. Potential societal risks, such as misuse for decision-making without proper validation, are acknowledged, and we emphasize that results should be interpreted with caution in high-stakes domains.

## REPRODUCIBILITY STATEMENT

We have taken multiple steps to ensure the reproducibility of our work. The proposed model, training procedures, and evaluation protocols are described in detail in the main text. Additional implementation details, including hyperparameter configurations, are provided in the appendix. All theoretical analyses are accompanied by complete proofs in the supplementary material. For datasets, we clearly describe the preprocessing steps and data split strategies in the supplementary document to facilitate re-implementation. To further support reproducibility, we submit anonymized source code and scripts as supplementary material, enabling independent verification of our results.

## LLM USAGE STATEMENT

We used large language models (LLMs) solely as auxiliary tools for improving writing clarity and refining grammar. The LLMs did not contribute to the conception of the research idea, algorithm design, experimental implementation, or analysis. All technical content, experiments, and conclusions were developed by the authors. The authors take full responsibility for the content of this paper.

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

## A    DATASETS DESCRIPTIONS

In this study, we utilize six diverse real-world datasets enriched with contextual features spanning various domains, including economics, health, web, stock markets, and natural sciences. Each dataset exhibits unique temporal characteristics and varying degrees of contextual dependency, offering a comprehensive benchmark.

- **Economic (FRED-MD)**: A monthly macroeconomic dataset consisting of 107 indicators across sectors such as production, labor, and inflation. It supports empirical studies requiring rich contextual interpretation.
- **Health (Covid-19)**: Released by Facebook's "Data for Good" initiative, this dataset tracks human mobility patterns across regions during the COVID-19 pandemic, offering policy-driven contextual signals.
- **Stock-NY (NYSE)**: Similar in structure and period to NASDAQ, this dataset provides daily time series from the New York Stock Exchange, facilitating comparative financial forecasting studies.
- **Stock-NA (NASDAQ)**: A daily stock dataset collected from the NASDAQ exchange between 2013 and 2017, containing representative securities with dynamics heavily influenced by external news and events.
- **Web (Wike2000)**: A high-dimensional dataset recording daily page views of 9,013 Wikipedia articles. We select the top 2,000 pages to capture volatile, event-driven user behavior shaped by external textual contexts.
- **Nature (CzeLan)**: A 30-minute resolution dataset capturing natural environmental signals with strong periodic patterns and low contextual dependence. It serves as a representative benchmark for low-context forecasting.

## B    ADDITIONAL IMPLEMENTATION DETAILS

In this appendix, we provide comprehensive descriptions of the baseline methods used for comparison in the main paper. We also detail the additional configuration parameters and training setups specific to our proposed model to ensure full reproducibility and transparency.

### B.1    COMPARED BASELINES

We first provide a detailed overview of the baseline models employed for comparative analysis in the main manuscript. These models are grouped into four distinct categories, each reflecting a key methodological paradigm in contemporary time series forecasting: LLM-based approaches, self-supervised frameworks, Transformer-based architectures, and a straightforward yet effective linear model. Below, we present concise descriptions of each model, emphasizing its core techniques and underlying conceptual foundations.

- **Time-LLM:** This is a reprogramming framework that transforms time series into text-based representations for input into a frozen large language model (LLM), guided by a Prompt-as-Prefix mechanism to enable reasoning and achieve general-purpose time series forecasting.
- **GPT4TS:** This work proposes the Frozen Pretrained Transformer (FPT), a framework that repurposes language or vision transformers for general time series analysis by freezing their core layers and fine-tuning only task-specific components, leveraging large-scale pretraining without requiring extensive time series data.
- **TimeDART:** This self-supervised pre-training framework addresses the challenge of modeling long-term dynamics and local patterns by combining Transformer encoding with a denoising diffusion process, yielding more transferable representations for downstream tasks.
- **SimMTM:** This is a masked time series pre-training framework that addresses the challenge of disrupted temporal semantics by reconstructing masked points through weighted aggregation from multiple complementary series, preserving temporal variations and learning manifold structures for improved downstream performance.

- **Autoformer:** This addresses the challenge of long-term time series forecasting by introducing a novel decomposition-based architecture with an Auto-Correlation mechanism, which replaces traditional self-attention to capture periodic dependencies and progressively model complex temporal patterns.

- **Crossformer:** This addresses the challenge of capturing temporal and inter-variable dependencies in multivariate time series forecasting using a Dimension-Segment-Wise embedding and Two-Stage Attention within a hierarchical encoder-decoder architecture.

- **DLinear:** This work challenges the effectiveness of complex Transformer-based models for long-term time series forecasting by demonstrating that a simple one-layer linear model can outperform them, highlighting limitations of self-attention in capturing temporal order and calling for renewed exploration of alternative approaches.

### B.2 MODEL CONFIGURATIONS

Next, we present the implementation details of our TokenCast framework, with a special focus on its three core stages: (a) time series discretization, (b) cross-modality alignment, and (c) generative fine-tuning. We design a specialized time series tokenizer to bridge structural differences across modalities. It consists of a causal TCN encoder that extracts contextualized embeddings and a causal Transformer decoder that reconstructs the original sequence. The embeddings are quantized into discrete tokens, producing compact and informative representations. Specifically, the encoder comprises 3 layers for effective feature extraction, with an embedding size of 64 and a uniform patch size of 4. The second stage aligns time series data with a pre-trained LLM by expanding its vocabulary to include time series tokens and introducing a unified projection layer for shared semantic space. The model is trained with an autoregressive objective using contextual features and historical tokens. Key hyperparameters, such as a learning rate of $5 \times 10^{-5}$ and batch size of 16, are carefully tuned to ensure stable alignment. For the final forecasting task, we utilize the aligned LLM in a generative manner. The model takes historical time series and relevant context as input to predict the sequence of future tokens. These generated tokens are then passed to the time series de-tokenizer to be converted back into a continuous predicted time series. For the optimization settings in this phase, we employ the Adam optimizer with a fine-tuning learning rate set to $1 \times 10^{-5}$. All parameters of the aligned model are updated during the fine-tuning process to adapt its generative capabilities specifically for multi-step horizon prediction, while retaining the same architectural configuration as the alignment phase.

## C FULL RESULTS

Due to space limitations, the complete results of all experiments are provided in the Appendix. The main experimental outcomes are summarized in Table 5, while the ablation studies on alignment and fine-tuning, as well as on multimodal contributions, are reported in Tables 2 and 3, respectively.

### C.1 FORECASTING PERFORMANCE ANALYSIS

Table 5 provides a comprehensive performance comparison across six benchmark datasets, evaluating models on both MSE and MAE metrics. Our model, TokenCast, demonstrates state-of-the-art performance, securing 17 first-place finishes and establishing itself as a top-tier method alongside the leading baseline. This aligns with findings that no single model universally excels, yet it highlights the advantages of our approach. Notably, other LLM-based baselines like Time-LLM and GPT4TS also deliver competitive results, which further validates the potential of leveraging large language models for time series forecasting. However, the performance of these models often varies significantly by dataset. For instance, while Time-LLM is highly effective on the Economic dataset, TokenCast shows a clear advantage on the Stock-NA benchmark, consistently outperforming all other models across nearly all forecasting horizons. This variability suggests that while powerful, generic LLM baselines may lack the specialized architecture needed to explicitly ground and adapt to diverse time-series dynamics. In stark contrast, earlier architectures like Crossformer and Autoformer consistently underperform, particularly on complex, non-stationary datasets such as Web and Economic. Their limitations are evident in the quantitative results; for example, on the Eco-

Table 5: All reported results are the average of three trials on various context-rich benchmark datasets. Lower values indicate better performance. Best results are in **bold** and second-best results are underlined.

| Model | | TokenCast | | Time-LLM | | GPT4TS | | TimeDART | | SimMTM | | Crossformer | | Autoformer | | DLinear | |
|---|---|---|---|---|---|---|---|---|---|---|---|---|---|---|---|---|---|
| Metric | | MSE | MAE | MSE | MAE | MSE | MAE | MSE | MAE | MSE | MAE | MSE | MAE | MSE | MAE | MSE | MAE |
| Economic | 24 | 38.946 | **1.188** | _38.235_ | 1.379 | 40.546 | 1.289 | **38.172** | 1.392 | 42.856 | _1.198_ | 386.326 | 3.892 | 66.721 | 1.714 | 69.693 | 1.621 |
| | 36 | **56.116** | **1.488** | _64.829_ | 1.626 | 67.560 | 1.569 | 66.131 | 1.632 | 70.291 | _1.511_ | 401.198 | 4.036 | 95.088 | 1.953 | 99.743 | 1.905 |
| | 48 | **77.678** | **1.767** | _92.481_ | 1.882 | 97.990 | 1.851 | 100.061 | 1.861 | 103.499 | _1.820_ | 415.148 | 4.156 | 130.566 | 2.195 | 137.429 | 2.220 |
| | 60 | **102.904** | **2.140** | _130.623_ | _2.153_ | 137.690 | 2.156 | 139.751 | 2.199 | 144.757 | 2.159 | 423.001 | 4.212 | 174.605 | 2.489 | 181.999 | 2.531 |
| | Avg | **68.911** | **1.646** | _81.542_ | 1.760 | 85.947 | 1.716 | 86.029 | 1.771 | 90.351 | _1.672_ | 406.418 | 4.074 | 116.745 | 2.088 | 122.216 | 2.069 |
| Health | 24 | **1.699** | _0.065_ | 2.322 | 0.081 | _1.961_ | **0.063** | 2.030 | 0.084 | 1.828 | 0.069 | 1644.541 | 2.363 | 2.097 | 0.276 | 24.717 | 0.426 |
| | 36 | **2.228** | **0.073** | 2.702 | 0.103 | _2.384_ | _0.080_ | 2.405 | 0.094 | 2.400 | 0.083 | 1645.156 | 2.536 | 2.407 | 0.256 | 29.370 | 0.463 |
| | 48 | **2.607** | _0.085_ | 2.916 | 0.114 | _2.710_ | 0.091 | 2.747 | **0.083** | 2.980 | 0.094 | 1645.237 | 2.588 | 2.770 | 0.253 | 16.203 | 0.381 |
| | 60 | 3.563 | 0.102 | 3.353 | 0.118 | _3.201_ | 0.096 | 3.309 | **0.092** | 3.672 | 0.107 | 1645.053 | 2.529 | **3.193** | 0.277 | 44.057 | 0.551 |
| | Avg | **2.524** | **0.081** | 2.823 | 0.104 | _2.564_ | _0.083_ | 2.623 | 0.088 | 2.720 | 0.088 | 1644.997 | 2.504 | 2.617 | 0.266 | 28.587 | 0.455 |
| Web | 24 | **453.609** | **1.169** | 524.512 | 1.699 | _510.602_ | 1.345 | 695.235 | _1.264_ | 1794.692 | 1.638 | 644.061 | 1.881 | 671.749 | 3.912 | 561.505 | 1.277 |
| | 36 | **480.019** | _1.204_ | 542.815 | 1.720 | 525.713 | 1.441 | 780.234 | 1.339 | _508.610_ | **1.162** | 706.661 | 1.930 | 736.403 | 3.302 | 630.056 | 1.362 |
| | 48 | **518.164** | _1.281_ | 571.482 | 1.783 | 551.430 | 1.504 | 779.730 | 1.398 | _531.377_ | **1.224** | 719.304 | 2.002 | 736.918 | 3.076 | 649.944 | 1.439 |
| | 60 | **537.828** | _1.330_ | 592.523 | 1.803 | 574.221 | 1.543 | 839.340 | 1.474 | _555.918_ | **1.283** | 732.002 | 2.040 | 774.696 | 2.921 | 687.700 | 1.514 |
| | Avg | **497.410** | **1.246** | 557.833 | 1.751 | _540.492_ | 1.458 | 773.635 | 1.369 | 847.649 | _1.327_ | 698.316 | 1.963 | 722.506 | 3.303 | 632.301 | 1.398 |
| Stock-NY | 24 | **0.289** | **0.350** | 0.351 | 0.383 | 0.342 | 0.380 | 0.499 | 0.477 | _0.332_ | _0.377_ | 1.100 | 0.909 | 0.427 | 0.466 | 0.600 | 0.573 |
| | 36 | **0.372** | **0.401** | 0.403 | 0.412 | 0.401 | 0.409 | 0.661 | 0.551 | _0.398_ | _0.406_ | 1.032 | 0.886 | 0.526 | 0.508 | 0.856 | 0.702 |
| | 48 | **0.538** | _0.479_ | 0.599 | 0.569 | 0.576 | 0.523 | 0.852 | 0.643 | _0.552_ | **0.477** | 1.061 | 0.889 | 0.757 | 0.605 | 1.112 | 0.802 |
| | 60 | **0.727** | **0.588** | 1.293 | _0.674_ | 1.232 | 0.697 | 1.092 | 0.753 | 1.170 | 0.719 | 1.251 | 0.962 | _0.994_ | 0.714 | 1.447 | 0.938 |
| | Avg | **0.482** | **0.455** | 0.662 | 0.510 | 0.638 | 0.502 | 0.776 | 0.606 | _0.613_ | _0.495_ | 1.111 | 0.912 | 0.676 | 0.573 | 1.004 | 0.754 |
| Stock-NA | 24 | **0.661** | **0.600** | _0.725_ | 0.700 | 0.796 | 0.689 | 0.796 | 0.711 | 0.867 | _0.677_ | 1.773 | 0.996 | 1.171 | 0.811 | 1.498 | 0.920 |
| | 36 | **0.887** | **0.694** | _0.962_ | 0.828 | 1.122 | 0.828 | 0.983 | _0.737_ | 1.281 | 0.827 | 1.921 | 1.046 | 1.494 | 0.912 | 1.755 | 0.974 |
| | 48 | _1.473_ | **0.886** | **1.385** | 1.004 | 1.481 | 0.956 | 1.822 | 1.028 | 1.577 | _0.907_ | 1.975 | 1.072 | 1.712 | 0.958 | 1.852 | 0.982 |
| | 60 | **1.515** | 0.941 | 1.729 | 1.166 | 1.688 | 1.045 | 2.036 | 1.055 | 1.646 | **0.924** | 1.980 | 1.095 | _1.608_ | _0.930_ | 1.733 | 0.955 |
| | Avg | **1.134** | **0.780** | _1.200_ | 0.925 | 1.272 | 0.880 | 1.409 | 0.883 | 1.343 | _0.834_ | 1.912 | 1.052 | 1.496 | 0.903 | 1.710 | 0.958 |
| Nature | 24 | _0.179_ | _0.246_ | 0.212 | 0.254 | 0.226 | 0.270 | **0.170** | **0.245** | 0.205 | 0.255 | 0.623 | 0.453 | 0.325 | 0.383 | 0.288 | 0.396 |
| | 48 | _0.252_ | 0.293 | **0.251** | _0.281_ | 0.271 | 0.303 | 0.261 | **0.271** | 0.262 | 0.295 | 0.724 | 0.513 | 0.556 | 0.509 | 0.364 | 0.448 |
| | 96 | 0.291 | 0.307 | _0.275_ | _0.293_ | 0.286 | 0.305 | **0.254** | **0.274** | 0.268 | _0.288_ | 0.734 | 0.516 | 0.502 | 0.487 | 0.354 | 0.423 |
| | 192 | 0.355 | 0.340 | _0.295_ | **0.302** | 0.311 | 0.318 | **0.286** | 0.313 | 0.301 | _0.304_ | 0.849 | 0.562 | 0.648 | 0.564 | 0.470 | 0.475 |
| | Avg | 0.269 | 0.297 | _0.258_ | _0.283_ | 0.274 | 0.299 | **0.243** | **0.276** | 0.259 | 0.286 | 0.733 | 0.511 | 0.508 | 0.486 | 0.369 | 0.436 |
| 1st Count | | **17** | **11** | 2 | 1 | 0 | 1 | 4 | 5 | 0 | 5 | 0 | 0 | 1 | 0 | 0 | 0 |

nomic dataset, the average MSE for Crossformer (423.001) and Autoformer (174.605) is substantially higher than that of TokenCast (68.911). This large performance gap underscores the difficulty their feature interaction mechanisms face in capturing intricate time-series patterns. In summary, TokenCast achieves not only state-of-the-art but also highly consistent results across a wide range of scenarios. We attribute this success to its core design: discretizing the time series into a unified token-based paradigm. By modeling time-series forecasting as a sequence-to-sequence task in this discrete space, TokenCast effectively captures the intricate dependencies and dynamics that challenge other methods, proving its reliability and effectiveness across diverse forecasting scenarios.

Table 6: Ablation study on the significant effects of cross-modality alignment and generative fine-tuning across multiple diverse datasets.

| Dataset | Economic | | Health | | Web | | Stock-NY | | Stock-NA | | Nature | |
|---|---|---|---|---|---|---|---|---|---|---|---|---|
| Metrics | MSE | MAE | MSE | MAE | MSE | MAE | MSE | MAE | MSE | MAE | MSE | MAE |
| w/ Cross-modality Alignment, w/o Generative Fine-tuning | 406.418 | 4.074 | 2.875 | 0.084 | 555.375 | 1.447 | 0.556 | 0.479 | 1.317 | 0.813 | 0.378 | 0.357 |
| w/o Cross-modality Alignment, w/ Generative Fine-Tuning | 72.292 | 1.690 | 2.783 | 0.079 | 504.740 | 1.264 | 0.515 | 0.478 | 1.181 | 0.804 | 0.305 | 0.318 |
| **w/ Cross-modality Alignment, w/ Generative Fine-tuning** | **68.961** | **1.695** | **2.524** | **0.081** | **497.410** | **1.246** | **0.482** | **0.455** | **1.134** | **0.780** | **0.269** | **0.297** |

## C.2 ALIGNMENT AND FINE-TUNING

We conduct the ablation study on two crucial training steps: the cross-modality alignment and generative fine-tuning. The comprehensive results in Table 6 clearly demonstrate their indispensable contribution to the overall framework performance. The model equipped with the cross-modality alignment stage consistently achieves lower MSE scores across all six datasets. Without this alignment, contextual features risk being misinterpreted by the time series backbone, leading to suboptimal forecasts. This highlights its critical role in effectively integrating contextual information by

bridging structural and semantic discrepancies between time series and contextual features, thus facilitating meaningful feature interaction. This alignment thus acts as a foundational step, ensuring the subsequent fine-tuning stage operates on a semantically rich and coherent feature space.

Concurrently, Table 6 vividly illustrates the pivotal contribution of the generative fine-tuning stage. Across all six benchmark datasets, the model employing generative fine-tuning consistently and substantially outperforms its counterpart that omits this crucial step. The performance degradation when omitting this stage is notable across various datasets, underscoring the general applicability and importance of the fine-tuning process. This drop is particularly stark on datasets like Stock-NA, where the complex, non-stationary patterns demand task-specific adaptation. Ultimately, these findings emphasize that generative fine-tuning is essential for adapting the pre-trained LLM's general capabilities to generative time series forecasting.

Table 7: Ablation study on the contribution of multimodal context (general info, local info, and text) across multiple diverse datasets.

| Dataset | Economic | | Health | | Web | | Stock-NY | | Stock-NA | | Nature | |
| Metrics | MSE | MAE | MSE | MAE | MSE | MAE | MSE | MAE | MSE | MAE | MSE | MAE |
|---|---|---|---|---|---|---|---|---|---|---|---|---|
| w/o Text | 80.000 | 1.820 | 2.950 | 0.089 | 530.000 | 1.320 | 0.550 | 0.490 | 1.300 | 0.820 | 0.312 | 0.325 |
| w/o General Info | 72.500 | 1.740 | 2.800 | 0.085 | 510.000 | 1.280 | 0.530 | 0.470 | 1.200 | 0.795 | 0.285 | 0.310 |
| w/o Local Info | 70.200 | 1.710 | 2.600 | 0.083 | 503.500 | 1.260 | 0.505 | 0.460 | 1.180 | 0.790 | 0.275 | 0.305 |
| **TokenCast (Ours)** | **68.961** | **1.695** | **2.524** | **0.081** | **497.410** | **1.246** | **0.482** | **0.455** | **1.134** | **0.780** | **0.269** | **0.297** |

## C.3 MUTIMODAL CONTRIBUTION

Table 7 presents the ablation study on the contribution of different multimodal contextual features, including general information, local information, and text. The results show that removing any type of contextual feature consistently degrades forecasting accuracy across all datasets, confirming that these sources of context play complementary roles in enhancing representation quality. In particular, discarding textual information (w/o Text) causes the most significant performance drop, especially on datasets with complex and non-stationary patterns such as Health and Nature, where domain knowledge and event-related semantics are crucial for interpreting abrupt changes and long-term shifts. This demonstrates the irreplaceable role of textual signals in providing external knowledge that cannot be inferred solely from numerical series. Meanwhile, removing general or local information also produces notable accuracy loss, which highlights their importance for aligning forecasts with contextual background (e.g., seasonal profiles, regional variations, or localized dynamics). These findings suggest that different types of contextual features contribute complementary perspectives: text offers semantic depth, general information provides global guidance, and local information ensures fine-grained adaptability.

Overall, TokenCast (Ours) achieves the best performance across all datasets, validating that the joint incorporation of textual, general, and local contextual features is essential for effective multimodal time series forecasting. This ablation analysis further emphasizes that weakening any single modality reduces the model's ability to capture the full spectrum of temporal dependencies, while integrating all contextual features leads to the most reliable and accurate forecasts.

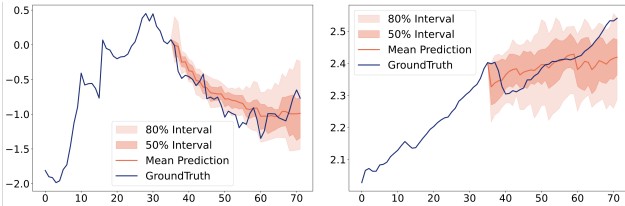

Figure 6: Forecasting with uncertainty on Stock-NY (left) and Economic (right) datasets. The plots compare the ground truth trajectories with the model's mean predictions, along with the 50% and 80% predictive intervals.

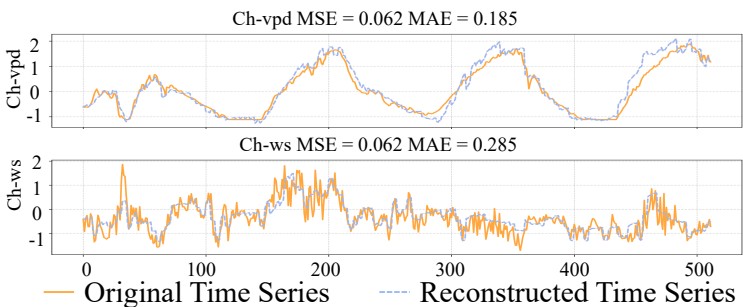

Figure 7: Visualizing the reconstruction of the Nature dataset in the vector quantized networks.

### C.4 GENERATIVE UNCERTAINTY

As shown in Figure 6, we evaluate the LLM's predictive uncertainty by performing multiple stochastic runs on the same input. The resulting forecasts, although varied across runs, form a coherent ensemble that consistently encompasses the ground truth. This behavior highlights the model's ability to represent meaningful uncertainty without deviating significantly from the actual data dynamics. Moreover, the forecasts' stability across different stochastic runs further underscores the predictive stability of our model, demonstrating that it can reliably capture uncertainty while maintaining high fidelity to the underlying data. This consistent performance reflects the model's stability, making it suitable for practical forecasting tasks where predictability and trustworthiness are essential. To validate the uncertainty modeling capabilities of our TokenCast, we conduct experiments on both the Economic and Stock-NY datasets. As shown in Figure 6, our method produces predictive distributions that closely track the ground truth, with 50% and 80% prediction intervals capturing the inherent variability in the data. By adjusting the temperature during sampling, we observe that the model can flexibly modulate the spread of the predictive intervals, indicating its potential for controllable uncertainty-aware forecasting. This demonstrates that our model not only provides accurate mean predictions but also yields well-calibrated uncertainty estimates.

### C.5 RECONSTRUCTION ANALYSIS OF TOKENIZER

Figure 7 presents reconstruction results on two representative channels of the Nature dataset, clearly illustrating the ability of our discretization module to generalize across time series with different levels of complexity. For Ch-vpd (top), the reconstructed sequence almost completely overlaps with the original series, yielding very low errors (MSE = 0.062, MAE = 0.185). This shows that the module preserves both global seasonal trends and fine-grained local fluctuations, ensuring that long-term periodic patterns are faithfully retained. For Ch-ws (bottom), the series exhibits greater variability and irregular spikes, posing a more challenging scenario. Nevertheless, the reconstructed sequence closely follows the underlying dynamics of the original data, with errors kept at controlled levels (MSE = 0.062, MAE = 0.285). The consistency between reconstructions and ground truth indicates that the tokenizer is not biased toward smooth series but adapts flexibly to noisy.

Overall, these results highlight two key strengths of our approach: stability, as reconstruction remains reliable across channels with distinct characteristics, and fidelity, as both trend-level and detail-level structures are preserved. Such properties are essential for downstream forecasting, where the quality of discretized representations directly determines predictive performance. By achieving accurate reconstructions on heterogeneous channels, our tokenizer provides a dependable basis for context-aware and domain-agnostic time series modeling.

### C.6 VISUALIZATION

Figure 8 presents a qualitative comparison of 36-to-36 forecasts on the Stock-NA dataset. The LLM-based models (TokenCast, Time-LLM, GPT4TS, and SimMTM) closely follow the ground truth, capturing major turning points and preserving key high-frequency variations. Although amplitude is not always exact, the directionality and regime changes are well tracked, which is critically important for financial time series. For the non-LLM baselines, behaviors diverge. Crossformer in this figure

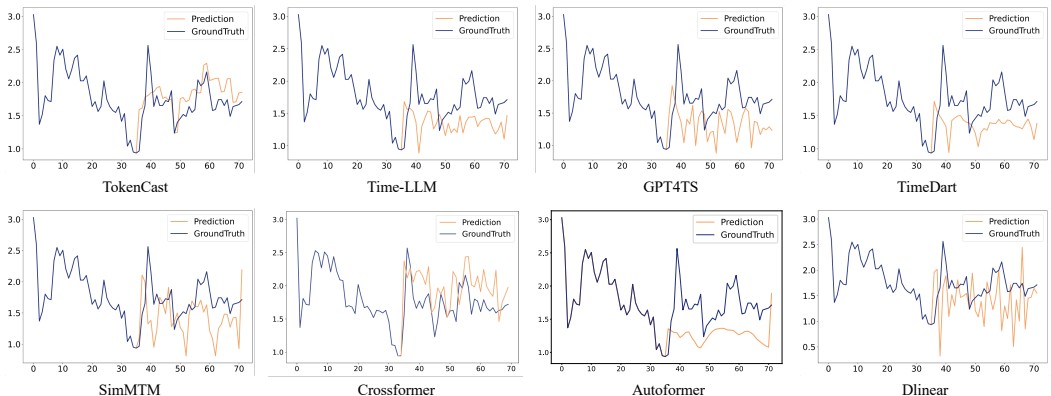

Figure 8: Visualize the 36-to-36 prediction results of different models on the Stock-NA dataset.

maintains reasonable alignment after the forecasting start, but shows damped amplitudes (variance shrinkage) and occasional phase lag around sharp moves (e.g., near the regime change around steps at 35–45 and subsequent fluctuations), leading to systematic underestimation of peaks and troughs rather than collapse. Autoformer tends to over-smooth, missing part of the local volatility. DLinear exhibits higher variance and noisy deviations, while TimeDart generally underestimates magnitudes and gradually drifts away from local fluctuations. Overall, the qualitative evidence indicates that LLM-based methods yield more coherent and responsive forecasts under the non-stationary, volatile conditions of stock data, whereas earlier architectures often suffer from amplitude underestimation, excessive smoothing bias, or noisy trajectories. This strongly supports the effectiveness of the unified token-based paradigm for capturing complex temporal dynamics.

