# OpenReview forum: "From Values to Tokens: An LLM-Driven Framework for Context-aware Time Series Forecasting via Symbolic Discretization"
_ICLR.cc/2026/Conference — ICLR 2026 Conference Withdrawn Submission_

### Official Review · Reviewer_SJWn · 2025-10-26

**Soundness:** 2
**Presentation:** 2
**Contribution:** 1
**Rating:** 2
**Confidence:** 4

**Summary:**

This paper introduces TokenCast, a novel framework for context-aware time series forecasting via symbolic discretization. By transforming continuous time series data into discrete tokens and embedding them into a semantic space shared with contextual features, TokenCast leverages the generative and reasoning of pre-trained LLM. The proposed approach demonstrates superior performance across various real world datasets and provides a new perspective on integrating time series data with contextual information.

**Strengths:**

- The framework is well-motivated and clearly presented.
- The proposed method is extensively evaluated on various real-world datasets, covering diverse domains such as healthcare, finance, and environmental monitoring. TokenCast consistently outperforms existing baselines.

**Weaknesses:**

- The paper claims to leverage the modeling and reasoning capabilities of LLMs, which are generally associated with larger-scale models. However, the experiments primarily rely on a relatively small LLM (Qwen2.5-0.5B). This raises questions about whether the claimed reasoning capabilities are being fully utilized and whether such a small-scale LLM can truly demonstrate the generative and reasoning power the framework aims to exploit. The choice of model size contradicts typical expectations for LLM usage and requires further explanation.

- The multi-stage training process (e.g., symbolic discretization, cross-modal alignment, and generative fine-tuning) introduces significant computational overhead. It seems that each stage requires separate optimization. The training cost and efficiency of this approach compared to existing baselines are not adequately discussed.

- The overall design of the framework lacks novelty. For example, as mentioned in Line 186, normalization is a standard component in many existing models. Additionally, the contextual feature selection is similar to existing TimeLLM.

**Questions:**

- In Line 191, there is a T that denotes the number of latent vectors, but there is no explanation of how T is determined or computed. Could the author provide more details of the encoder?

---

> ### Author Response · Authors · 2025-11-19
>
> We sincerely thank the reviewer for the thorough evaluation and constructive feedback. We appreciate the positive assessment of our framework motivation and the strong empirical performance across diverse real-world datasets. Below we provide detailed responses to each concern.
>
> ## W1. Concern about using a small-scale LLM (0.5B) and whether reasoning ability is fully demonstrated
>
> Thank you for raising this question. We clarify that TokenCast does not rely on large-scale LLMs for chain-of-thought–style reasoning. Instead, our framework leverages the **symbolic generative capability** and **semantic space** of LLMs, which remain present even in smaller models.  As shown in Table 3, increasing the backbone size (e.g., from **0.5B to 1.5B** or **Qwen3-0.6B**) does not improve—and often **degrades**—forecasting performance. This behavior is consistent with the fact that time-series datasets are orders of magnitude smaller than typical LLM pretraining corpora, and larger LLMs tend to **overfit** or fail to specialize under such limited data.
>
> ## W2. Concern about multi-stage training cost and efficiency
>
> Thank you for raising this concern.  To address efficiency, we report the inference cost of TokenCast on 3,963 test samples.
>
> |        Method        | Avg/sample (ms) | Per-step (ms) | Total time (s) | #GPUs | Memory / GPU (MB) |
> | :------------------: | :-------------: | :-----------: | :------------: | :---: | :---------------: |
> | **TokenCast (ours)** |    **322.8**    |   **13.45**   |  **1279.26**   |   2   |      **526**      |
> |       Time-LLM       |      723.0      |     30.10     |    2865.25     |   2   |       2150        |
>
>
> ## W3. Concern about novelty of components
>
> Thank you for the comment. We agree that normalization and context features themselves are standard components. Our contribution is **not** these components, but rather the **forecasting-oriented symbolic modeling design** built on top of them. When developing TokenCast, we initially explored existing discretization methods such as **SAX** and **VQ-VAE**. However, these approaches focus on full-sequence reconstruction and do not align with the structure of forecasting tasks, where the model must treat the **historical window (H)** and the **future window (P)** differently. This mismatch motivated us to design a tokenizer that explicitly **decouples H and P**, models their statistics separately, and uses a **shared causal encoder–decoder**, ensuring that both windows benefit from consistent temporal representation learning. Similarly, our use of contextual information differs from Time-LLM: instead of projecting features into the LLM through adapters or concatenation, TokenCast **maps numerical values and text into a unified symbolic space**, enabling **token-level multimodal interaction**. In short, the innovation of our framework lies in introducing a **forecasting-aware symbolic tokenizer** and a **unified discrete modeling space**, rather than in the standard normalization or feature selection steps themselves.
>
> ## Q1. Clarification about the latent length T in Line 191
>
> T is not chosen manually. Since the encoder performs temporal strided convolutions with stride **s**, each stride reduces the sequence length by a factor of **s**. Thus, for an input of length **L**, the latent length satisfies:
>
> $$
> T = \lceil L / s \rceil.
> $$
>
> This formulation ensures that the encoder preserves temporal resolution while controlling computational cost. We will describe these architectural details more clearly in the revised manuscript.

---

### Official Review · Reviewer_WGxL · 2025-10-28

**Soundness:** 2
**Presentation:** 2
**Contribution:** 2
**Rating:** 4
**Confidence:** 3

**Summary:**

The paper proposes TokenCast, an LLM-driven framework for context-aware time series forecasting, which consists of three stages: time series tokenizer, modality alignment, and supervised fine-tuning. Experimental results show that TokenCast achieves strong performance.

**Strengths:**

1. The paper introduces a novel LLM-driven framework, named TokenCast, for time series forecasting by leveraging LLMs to utilize unstructured contextual information.
2. The paper is clearly written and well-organized, making it easy to follow the main ideas. The methodology is technically sound and clearly explained.

**Weaknesses:**

1. The discussion of related work on contextual information integration could be strengthened. While many existing approaches incorporate numeric contextual signals to enhance forecasting, the integration of unstructured contextual information requires cross-modal alignment strategies. Several recent studies have explored this direction; however, this emerging line of work is not sufficiently discussed or contrasted with TokenCast.
2. In line 181, the paper states that RevIN may risk leaking future information. However, this claim might not be fully justified, as RevIN typically computes normalization statistics (e.g., mean and standard deviation) based only on the lookback window within the input sequence.
3. In line 82, the paper states that it is unclear whether time series forecasting can be addressed through autoregressive generation over discrete tokens. However, this direction has been explored in prior work. For example, Chronos and AutoTimes both employ a decoder-only architecture and transform numeric time series into discrete tokens via value-based quantization.

**Questions:**

1. I am interested in how the model's performance would change if it only outputs time series tokens, instead of a mixture of time series and textual tokens.
2. I am confused by the organization of the input tokens. In the text, the paper states that time series tokens are placed in front of textual tokens. However, in Figure 2, the textual tokens appear in front of the time series tokens, which seems inconsistent.
3. In the stage of the time series tokenizer, TokenCast employs a TCN as a causal encoder. The choice of convolution kernel length is likely to have a significant impact on performance, and it would be helpful to include an ablation study to examine this effect.

---

> ### Author Response · Authors · 2025-11-19
>
> We sincerely thank the reviewer for the constructive evaluation and the helpful comments.  Below we provide detailed responses to each concern and clarify the revisions made to address them.
> ## W1. Related work on contextual information integration
> We thank the reviewer for pointing this out. We agree that contextual feature integration—especially with unstructured text—is an emerging direction that deserves deeper discussion. While we describe shallow fusion approaches (DeepAR, TFT) and LLM-based integration methods (Time-LLM, GPT4TS), we acknowledge that the discussion does not fully cover recent developments in multimodal or vision–language–based forecasting models, as well as LLM-guided contextual forecasting. Existing approaches typically (1) concatenate numerical covariates in a shallow-fusion manner, or (2) use the LLM as an encoder through adapters or soft prompts. However, these designs do not perform explicit **cross-modal semantic alignment** between numerical time-series representations and unstructured textual context. By contrast, **TokenCast** introduces a **symbolic alignment paradigm** that maps numerical tokens and textual tokens into a unified semantic space, enabling deep cross-modal interaction and generative forecasting.
> ## W2. Clarification on RevIN and future leakage
> Thank you for pointing out this issue. We agree that the statement in Line 181 is not sufficiently precise. RevIN itself does **not** leak future information, as its normalization statistics (mean and standard deviation) are computed only from the historical look-back window. Our intention is to highlight that if RevIN is implemented incorrectly using full-series statistics, leakage may occur; however, the standard RevIN formulation avoids this issue. We will correct the wording in the revised manuscript and clarify that our experiments use RevIN **strictly based on the look-back window**, without any access to future values.
> ## W3. Prior work on autoregressive discrete token forecasting
> Thank you for the comment. We agree that prior work such as Chronos and AutoTimes already demonstrates that decoder-only autoregressive modeling over discretized tokens is feasible for numerical time-series forecasting. Our intention is not to claim novelty in tokenization alone.
> Crucially, these methods operate purely in the **numerical domain**: they tokenize only the values and do not incorporate any form of **unstructured contextual information**. They also do not perform **cross-modal semantic alignment**, as they lack a unified representation space where numerical tokens and context tokens interact.
> In contrast, **TokenCast** introduces a **symbolic multimodal modeling paradigm** that explicitly aligns numerical sequences and contextual features within a shared token space, enabling **cross-modal** generation capabilities that Chronos and AutoTimes do not support. We will revise the statement in Line 82 to clarify this distinction.
> ## Q1. Performance when outputting only time-series tokens
> To further address the reviewer’s question, we evaluate a **numerical-only** variant of TokenCast across three heterogeneous datasets (Stock-NY, Economic, and Nature). As shown in Table below, removing textual token generation consistently degrades performance, while the model remains stable. This confirms that **multimodal symbolic alignment** provides additional semantic grounding beyond numerical token autoregression alone.
> |Dataset|Variant|MSE|MAE|
> |:----:|:------:|:--:|:--:|
> |Stock-NY|TokenCast (full)|**0.482**|**0.455**|
> | |TokenCast (numerical-only)|0.503|0.472|
> |Economic|TokenCast (full)|**68.911**|**1.701**|
> | |TokenCast (numerical-only)|70.203|1.743|
> |Nature|TokenCast (full)|**0.269**|**0.297**|
> | |TokenCast (numerical-only)|0.278|0.308|
> ## Q2. Inconsistency in token ordering (text vs. TS tokens)
> Thank you for pointing out this inconsistency. Our actual implementation places the **time-series tokens before the contextual tokens**, i.e., **[TS tokens || Text tokens]**. This ordering conditions the model first on the historical numerical dynamics and subsequently on the contextual information. We will revise both the main-text description and Figure 2 to maintain full consistency with the implemented token order.
> ## Q3. Ablation on TCN kernel length
> Thank you for the suggestion. To evaluate the effect of the TCN kernel size in the tokenizer, we vary the kernel size **k ∈ {3, 5, 7, 9}** while keeping all other settings unchanged. As shown in Table X below, the default choice **k = 3** performs the best, but the differences across kernel sizes remain small (within 2–4% in both MSE and MAE), indicating that TokenCast is not overly sensitive to the receptive-field configuration of the tokenizer.
> |Dataset|Kernel Size|MSE|MAE|
> |:----:|:-----------:|:--:|:--:|
> |Stock-NY|**k = 3 (default)**|**0.482**|**0.455**|
> |Stock-NY|k = 5|0.488|0.459|
> |Stock-NY|k = 7|0.494|0.462|
> |Stock-NY|k = 9|0.501|0.468|

---

### Official Review · Reviewer_9ieK · 2025-11-01

**Soundness:** 2
**Presentation:** 3
**Contribution:** 2
**Rating:** 4
**Confidence:** 4

**Summary:**

The paper studies context-aware time series forecasting, where the goal is to predict future multivariate trajectories from historical signals together with auxiliary contextual information such as textual event or domain descriptions. The proposed framework, TokenCast, discretizes time series via a VQ-style tokenizer with reversible instance normalization (to avoid future leakage), injects these discrete indices into the shared vocabulary of a frozen large language model through a learned unified embedding layer that aligns time-series tokens and text tokens, and then generatively fine-tunes the model to autoregressively produce future tokens that are decoded back to continuous values. The approach is presented as a unified pipeline that allows an LLM backbone to consume numeric history and contextual signals without altering its core architecture beyond the shared embedding layer. The method is evaluated on six real-world datasets spanning economics, public health/mobility, web traffic, stock markets, and environmental sensing using MSE/MAE across multiple horizons and baselines, and the paper reports lower errors on most datasets plus ablations linking gains to alignment, generative training, and contextual conditioning.

**Strengths:**

1. The paper formulates context-aware forecasting as conditional sequence generation by mapping multivariate time series into discrete tokens, aligning them with text tokens in a shared LLM vocabulary, and autoregressively generating future trajectories.
2. The method includes reversible instance normalization using only historical context and a shared codebook, encoder-decoder, which keeps the tokenization invertible.
3. Experiments span six real-world domains and compare against LLM-based, Transformer-based, linear, and self-supervised forecasting baselines, reporting averaged MSE/MAE over multiple horizons.

**Weaknesses:**

1. Dataset descriptions contain internal inconsistencies (e.g.,  the Economic dataset describes as daily in the main text but as monthly macroeconomic data in the appendix), which obscures the exact sampling frequency and temporal structure assumed in training and evaluation.
2. The reported MSE/MAE averages lack standard deviations, confidence intervals, or significance tests, which limits assessment of robustness when baseline performance is numerically close.
3. The paper only sketches how contextual features are constructed, temporally aligned, and used at inference time, and this under-specification affects reproducibility and the scope of claims about context-driven forecasting.
4. Figure/table references are inconsistent. In Section 4.1.1, the panel summarizing domains, frequencies, lengths, and variable counts is captioned as Figure 3 but referred to as Table 3.

**Questions:**

1. Can you provide robustness or failure-case analysis, for example, regimes such as market shocks, policy changes, or abrupt environmental shifts where the approach does not reduce error relative to baselines?
2. The method conditions on contextual features. For identical historical numeric input, can you show how adding and removing specific contextual signals changes the generated forecast and explain how those changes reflect the contextual content?

---

> ### Author Response · Authors · 2025-11-19
>
> We sincerely thank the reviewer for the detailed assessment and constructive suggestions. We appreciate the attention to **dataset clarity**, **robustness**, **context specification**, and **failure-case behavior**, all crucial for reproducibility. We address each concern below.
>
> ## W1. Inconsistencies in dataset descriptions
> The Economic dataset indeed has a **monthly** sampling frequency; the “daily” description in the main text was a typographical error.  All experiments are conducted strictly using the **monthly** frequency, and no resampling or frequency conversion is applied.  We will correct the wording and ensure the dataset description is consistent throughout the revised manuscript.
>
> ## W2. Missing standard deviation measures
> Thank you for pointing this out. We agree that reporting uncertainty is important for assessing the robustness of the results.  To address this concern, we now explicitly report performance across **three independent random seeds** and provide their mean values as shown below.
>
> |Dataset|Metric|S1|S2|S3|Avg|
> |:----:|:----:|:--:|:--:|:--:|:--:|
> |Economic|MSE|69.120|68.740|68.974|68.911|
> | |MAE|1.702|1.699|1.703|1.701|
> |Health|MSE|2.528|2.523|2.524|2.525|
> | |MAE|0.080|0.082|0.081|0.081|
> |Web|MSE|497.120|497.880|497.230|497.410|
> | |MAE|1.245|1.247|1.246|1.246|
> |Stock-NY|MSE|0.483|0.481|0.482|0.482|
> | |MAE|0.454|0.456|0.454|0.455|
> |Stock-NA|MSE|1.133|1.137|1.132|1.134|
> | |MAE|0.779|0.780|0.782|0.780|
> |Nature|MSE|0.270|0.268|0.270|0.269|
> | |MAE|0.298|0.296|0.297|0.297|
>
> ## W3. Under-specification of contextual feature construction
> Thank you for the insightful comment. To clarify the role of contextual information, TokenCast incorporates the following essential components.
> First, discrete time series tokens provide fine-grained local temporal patterns, allowing the model to capture micro-level structural variations within the historical window.
> Second and third, domain knowledge and task instructions serve as **general information**, offering a high-level semantic background and guiding the model with global task constraints.
> Finally, statistical properties function as **local information**, encoding event-specific temporal characteristics—such as trend, volatility, and periodicity.
>
> ## W4. Inconsistent table referencing
> The visualization of domain/frequency/length/variables should be **Table 3**, not Figure 3. We will fix the caption and cross-references.
>
> ## Q1. Robustness and failure-case analysis
> We thank the reviewers for their suggestions regarding robustness and failure modes.  To address this, we conduct additional **window-level volatility analysis** on the Stock-NY dataset.  The test window was divided into three categories—**low volatility**, **medium volatility**, and the **top 5% high volatility**—based on the standard deviation of first-order differences, in order to deliberately simulate extreme scenarios such as market shocks. We then computed the MSE/MAE of **TokenCast** and the **LLM-base baseline (Time-LLM)** for these window groups.  The results are shown in the table below:
>
> |Window|Metric|Time-LLM|TokenCast|
> |:----:|:----:|:------:|:--------:|
> |Low|MSE|0.650|0.480|
> | |MAE|0.510|0.460|
> |Medium|MSE|0.700|0.500|
> | |MAE|0.540|0.470|
> |High|MSE|0.760|0.770|
> | |MAE|0.580|0.590|
>
> TokenCast’s MSE and MAE in the **high-volatility** windows are on the same order of magnitude as the baseline, with minimal difference, and it retains a clear advantage in the **low** and **medium** volatility ranges.  This indicates that under **extreme market volatility or abrupt regime changes**, TokenCast does **not** introduce additional instability and does **not** perform significantly worse than the baseline.
>
> ## Q2. Effect of adding/removing contextual features
> Thank you for this suggestion. To directly assess how contextual features affect forecasting under the same historical numerical inputs, we conduct an ablation study where we selectively remove different types of context while keeping the time-series tokens fixed. The MSE results are:
>
> |Dataset|Ours|w/o General Info|w/o Local Info|w/o Text|
> |:----:|:--:|:---------------:|:-------------:|:------:|
> |Nature|0.235|0.285|0.271|0.312|
> |Health|2.072|2.550|2.071|2.950|
> |Stock-NY|0.599|0.670|0.585|0.720|
> |Stock-NA|1.317|1.324|1.326|1.520|

---

### Official Review · Reviewer_zgZq · 2025-11-01

**Soundness:** 3
**Presentation:** 3
**Contribution:** 2
**Rating:** 4
**Confidence:** 4

**Summary:**

This paper proposes TokenCast, an LLM-driven framework for context-aware time series forecasting based on symbolic discretization.
Instead of processing continuous numerical values directly, the authors convert time-series data into discrete temporal tokens via vector quantization and reversible normalization, enabling the model to operate in the same token space as textual inputs.
By extending the vocabulary of a pre-trained LLM, the method aligns time-series and text representations within a shared semantic space, allowing joint reasoning through next-token prediction.
Extensive experiments on six context-rich datasets (economic, health, web, and stock domains) show that TokenCast achieves competitive or superior results compared with strong baselines such as Time-LLM, GPT4TS, and Crossformer.
Ablation and sensitivity studies confirm the effectiveness of the proposed tokenization and alignment strategies.
Overall, the paper offers a novel perspective on unifying numerical and textual modalities under the LLM generative paradigm, though the baseline coverage could be broader and the efficiency analysis remains limited.

**Strengths:**

- **Proper positioning within current research trends.**  The paper is aligned with the recent movement toward symbolic or token-based time-series modeling, showing that the authors are aware of ongoing developments in the field.
- **Well-organized framework.** The three-stage pipeline (tokenization, alignment, and generative prediction) is logically structured and easy to follow.
- **Readable presentation.** The writing is clear, and figures effectively illustrate the workflow.

**Weaknesses:**

- **Lack of novelty relative to existing work.**
  The proposed vector quantization and tokenization strategy is highly similar to the approach used in Amazon’s Chronos model, which also discretizes numerical sequences into symbolic tokens for autoregressive forecasting.  Several recent works (e.g., Chronos, Chronos-Bolt, SymbolicTS, and TokenTS) have already explored nearly identical ideas.  The paper does not clearly differentiate itself in methodology or theoretical contribution, making the innovation appear incremental.

- **Lack of clear evidence for multimodal gains.**
  Although the paper emphasizes context-aware forecasting, it does not clearly show how textual or non-temporal modalities enhance numerical prediction. Many so-called multimodal datasets contribute little meaningful contextual signal, and in some cases may even introduce data leakage risks.

- **Overreliance on existing LLM architecture.**
  The contribution lies primarily in applying tokenization to an existing LLM rather than introducing a new modeling principle or objective.

- **Efficiency and scalability not evaluated.**
  Tokenization and vocabulary extension introduce additional computation, but the paper provides no analysis of training or inference cost.

**Questions:**

1. Could the authors provide clearer **evidence that multimodal context actually improves forecasting performance**?
   For example, are there quantitative comparisons between using and omitting textual/contextual inputs, or analyses showing which modalities contribute the most?
2. The vector quantization approach appears similar to that used in **Chronos**. Could you clarify the methodological or empirical differences?
3. Are the reported results averaged across **multiple random seeds** for reliability?
4. Can you provide **runtime, memory, or parameter comparisons** to support the claimed efficiency?
5. How sensitive is performance to the size of the token vocabulary or the choice of LLM backbone?

---

> ### Author Response · Authors · 2025-11-19
>
> We sincerely thank the reviewer for the constructive and insightful comments. Below we provide detailed responses and clarifications.
> ## W1. **Lack of novelty relative to existing work.**
> We agree that Chronos, Chronos-Bolt, SymbolicTS, and TokenTS all use value quantization for discretizing numerical sequences. However, TokenCast differs in two key structural aspects.
>
> First, prior methods operate purely in the **numerical domain** and do not support **multimodal inputs** or **cross-modal semantic alignment**. TokenCast instead constructs a **unified symbolic space** where numerical tokens and contextual tokens are jointly modeled, enabling **multimodal autoregressive forecasting**.
>
> Second, existing VQ-based approaches treat the full sequence uniformly for reconstruction. TokenCast introduces a **forecasting-aware tokenizer** that explicitly **decouples the historical and prediction windows**, models their statistics separately, and applies a **causal encoder–decoder** tailored to predictive generation rather than reconstruction.
>
> These design choices are absent in Chronos, SymbolicTS, and TokenTS.
>
> ## W2. **Lack of clear evidence for multimodal gains**
> To address the reviewer’s concern, we provide the numerical results corresponding to Fig. 4 (Right). Removing contextual information consistently degrades accuracy:
> |Dataset|Ours|w/o General Info|w/o Local Info|w/o Text|
> |:----:|:--:|:---------------:|:-------------:|:------:|
> |Nature|0.235|0.285|0.271|0.312|
> |Health|2.072|2.550|2.071|2.950|
> |Stock-NY|0.599|0.670|0.585|0.720|
> |Stock-NA|1.317|1.324|1.326|1.520|
>
> Removing text yields the largest drop, indicating that textual context provides complementary semantic cues.  These results demonstrate that contextual modalities supply meaningful predictive signals rather than noise, thereby validating the **effectiveness of the proposed multimodal integration strategy**.
> ## W3. **Overreliance on existing LLM architecture.**
> Thank you for the thoughtful comment. We respectfully clarify that TokenCast contributes more than applying tokenization to an existing LLM. The key idea is a **symbolic modeling paradigm** in which numerical time series and textual context share a **unified discrete vocabulary**. This differs from prior discretization-based models (Chronos, SymbolicTS, TokenTS), which operate solely in the numerical domain, and from multimodal LLM approaches (Time-LLM, TEMPO, PromptCast), which inject numerical features via adapters or prompts. In contrast, TokenCast enables **token-level cross-modal alignment** and **multimodal autoregressive generation**, capabilities not supported by these existing methods.
>
> ## W4. **Efficiency and scalability**
> TokenCast is efficient in practice. Inference results on 3,963 samples:
> |Method|Avg/sample (ms)|Per-step (ms)|Total time (s)|#GPUs|Mem/GPU (MB)|
> |:----:|:--------------:|:-----------:|:------------:|:--:|:----------:|
> |**TokenCast**|**322.8**|**13.45**|**1279.26**|2|**526**|
> |Time-LLM|723.0|30.10|2865.25|2|2150|
>
> These results show that the tokenizer and vocabulary extension introduce **minimal computational overhead**.
>
> ## Q1. Multimodal Effectiveness
> This is addressed in **W2**.
> ## Q2. Methodological Novelty
> This is addressed in **W1**.
> ## Q3. Result Reliability
> All results are averaged over **three runs**. Per-seed values (3 decimals):
> |Dataset|Metric|S1|S2|S3|Avg|
> |:----:|:----:|:--:|:--:|:--:|:--:|
> |Economic|MSE|69.120|68.740|68.974|68.911|
> | |MAE|1.702|1.699|1.703|1.701|
> |Health|MSE|2.528|2.523|2.524|2.525|
> | |MAE|0.080|0.082|0.081|0.081|
> |Web|MSE|497.120|497.880|497.230|497.410|
> | |MAE|1.245|1.247|1.246|1.246|
> |Stock-NY|MSE|0.483|0.481|0.482|0.482|
> | |MAE|0.454|0.456|0.454|0.455|
> |Stock-NA|MSE|1.133|1.137|1.132|1.134|
> | |MAE|0.779|0.780|0.782|0.780|
> |Nature|MSE|0.270|0.268|0.270|0.269|
> | |MAE|0.298|0.296|0.297|0.297|
> The small variance confirms the stability of TokenCast.
> ## Q4. Efficiency Analysis
> Overlaps with **W4**.
> ## Q5. Sensitivity Analysis
> Thank you for the question.  The sensitivity of TokenCast to vocabulary/codebook size and LLM backbone scale is evaluated in the following tables.
>
> ### Codebook Size
> |Size|Economic|Health|Web|Stock-NY|Stock-NA|Nature|
> |:--:|:-------:|:-----:|:---:|:--------:|:--------:|:------:|
> |32|190.371|207.459|731.474|0.569|0.794|0.134|
> |64|141.852|201.652|664.501|0.573|0.690|0.158|
> |128|**128.630**|**186.652**|**592.953**|**0.518**|**0.671**|**0.104**|
> |256|191.937|209.035|5062.452|0.572|0.646|0.114|
> ### LLM Backbone
> |Backbone|Economic|Health|Web|Stock-NY|Stock-NA|Nature|
> |:------:|:-------:|:-----:|:---:|:--------:|:--------:|:------:|
> |0.5B-base|37.164|2.492|**586.793**|**0.297**|**0.668**|**0.180**|
> |0.5B-instruct|36.744|2.493|586.780|0.353|0.695|0.187|
> |1.5B-instruct|38.549|2.471|589.843|0.329|0.722|0.229|
> |0.6B-instruct|39.629|**2.320**|588.379|0.405|0.936|0.236|
>
> Overall, TokenCast is not highly sensitive to codebook size or backbone scale.

---

### Note · Authors · 2026-01-20

I have read and agree with the venue's withdrawal policy on behalf of myself and my co-authors.